# Calculation and realization of new method grey residual error correction model

**Lifang Xiao** *, **Xiangyang Chen, Hao Wang**

School of Computer Science and Engneering, Wuhan Institute Technology, Wuhan, China

* 479448392@qq.com

## Abstract

Aiming at the problem of prediction accuracy of stochastic volatility series, this paper proposes a method to optimize the grey model(GM(1,1)) from the perspective of residual error. In this study, a new fitting method is firstly used, which combines the wavelet function basis and the least square method to fit the residual data of the true value and the predicted value of the grey model(GM(1,1)). The residual prediction function is constructed by using the fitting method. Then, the prediction function of the grey model(GM(1,1)) is modified by the residual prediction function. Finally, an example of the wavelet residual-corrected grey prediction model (WGM) is obtained. The test results show that the fitting accuracy of the wavelet residual-corrected grey prediction model has irreplaceable advantages.

**Data Availability Statement:** All relevant data are within the manuscript and its Supporting information files.

**Funding:** The authors received no specific funding for this work.

## 1. Introduction

Since its birth in 1982, the grey system theory has been widely used in the fields of economy, management and engineering technology [1–10]. Among them, the grey prediction model (GM(1,1)) is one of the core contents of the grey system theory. Its characteristics are that it needs fewer samples and the calculation is simple, so it is better than the traditional prediction method. However, in recent years, many scholars have proposed the improvement of the grey model and the study of its application scope The research can be roughly divided into two categories: 1) using the method of function transformation to improve the smoothness of the sequence can reduce the level ratio of the original data and make the level ratio fall within the allowable interval to achieve the purpose of improving the modeling accuracy; 2) According to the characteristics of the original data, we can further improve the gray prediction algorithm, optimize the background value, optimize the initial conditions, etc.

In literature [11–17], some scholars studied the function transformation from the perspective of the monotonicity of the function and the compression transformation, in order to further improve the accuracy of the model, and some scholars used the buffering operator to reduce the fluctuation. However, due to the need to restore the sequence of the new sequence generated by the function transformation after modeling, the accuracy of the final restoration is reduced and the effect is not ideal In literature [18–27], scholars not only believe that the average value of background value has errors, but also believe that it is unreasonable that the initial condition must pass through a certain point in the original data. Therefore, they propose

**Competing interests:** The authors have declared that no competing interests exist.

some methods to solve this problem, such as improving the background value and initial condition, so as to improve the accuracy of the model However, in the process of improving the background value, the convergence of iterations and in the process of constructing the optimal background value, we should consider how the convergence of the iterative formula. If the iterative formula doesn't converge, then we can't get the optimal background value.

However, for the stochastic volatility series, from the perspective of improving the smoothness, the background value and initial value of the sequence by means of function transformation, the final effect of improving GM(1,1) model is not good. Literature [28–30] studied the stochastic volatility series. In literature [20], accelerated translation transformation and weighted mean transformation are proposed to process the original data to make the original data smoothly change. The grey model(GM(1,1)) is established for the transformed smooth data to first predict, and then the corresponding inverse transformation is carried out for the predicted data, so as to improve the prediction accuracy of the grey model. Literature [29, 30] uses Fourier series to fit the residual and actual value of the mean model, and establishes the Fourier residual corrected grey prediction model (FGM) to predict practical problems, and the final prediction accuracy and effect was achieved good results.

In view of the stochastic volatility sequence prediction accuracy problem, this study put forward a kind of optimization from the perspective of residual method of grey GM (1, 1) model. Data fitting method is often used by people to obtain the approximate function relation, so as to provide quantitative relationship for further research. In many cases, the fitting function can be used to explain the data, in order to obtain better results than the interpolation and approximation function. The common method of data fitting is the least square method, and the basis functions mostly use n-degree polynomials, Chebyshev polynomials, Bernstein polynomials, trigonometric function [29, 30], etc., The different basis functions will produce different fitting effects, so we make different choices. However, the stability of the calculation process and the accuracy of fitting are the main criteria to evaluate the merits and demerits of the basic functions. In this study, the combination of wavelet function basis and least square method is used to fit the residual data of the grey model. Since the residual data presents the stochastic volatility state and the wavelet function has good local characteristics and fluctuation, the wavelet function is selected as the basis function, which has a good fitting effect for the stochastic volatility residual. The constructed residual prediction function can accurately reflect the variation of residual, and then the prediction function of average gray GM (1,1) model can be modified by the residual prediction function, so as to improve the fitting accuracy of stochastic volatility sequence. The WGM model in this study and the FGM model in the literature [29] used to determine the PetroChina stock price closing price on May 9, 2018 solstice on May 23 [31] and the number of tourists in Taiwan on August 1, 2006 solstice on July31, 2007 show that the fitting accuracy and fitting effect of the grey prediction model with wavelet residual correction are better than GM (1,1) and FGM. The precision of the grey model series is improved, and the application range of the grey model is broadened.

## 2. Methodology

### 2.1 Mean GM(1,1) model

The initial data is listed as $X^{(0)} = \{x^{(0)}(1), x^{(0)}(2), \cdots, x^{(0)}(n)\}$, the generated 1-AGO sequence is $X^{(1)} = \{x^{(1)}(1), x^{(1)}(2), \cdots, x^{(1)}(n)\}$ and $X^{(1)}(k) = \sum_{i=1}^{k} x^{(0)}(i)$. The first-order differential equation of the GM(1,1) model is

$$\frac{dx^{(1)}}{dt} + ax^{(1)} = b \tag{1}$$

Where t denotes the indepent variables in the system, a represents the developed coeffficient, b is the grey controlled variable, and a and b denote the model parameters requiring determination. The values of $a$ and $b$ become by the ordinal least-square method as

$$\begin{bmatrix} a \\ b \end{bmatrix} = (B^T B)^{-1} B^T Y \tag{2}$$

Furthermore, accumulated matrix $B$ is

$$B = \begin{bmatrix} -z^{(1)}(2) & -z^{(1)}(3) & \cdots & -z^{(1)}(n) \\ 1 & 1 & \cdots & 1 \end{bmatrix}^T,$$

$$z^{(1)}(k) = \frac{1}{2}[x^{(1)}(k) + x^{(1)}(k-1)], k = 1, 2, \cdots, n.$$

Meanwhile, the constant vector Y is

$$Y = \begin{bmatrix} x^{(0)}(2) & x^{(0)}(3) & \cdots & x^{(0)}(n) \end{bmatrix}^T$$

According to Eq (2), we have

$$a = \frac{\sum_{k=2}^{n} z^{(1)}(k) \sum_{k=2}^{n} x^{(0)}(k) - (n-1) \sum_{k=2}^{n} z^{(1)}(k) x^{(0)}(k)}{(n-1) \sum_{k=2}^{n} [z^{(1)}(k)]^2 - [\sum_{k=2}^{n} z^{(1)}(k)]^2} \tag{3}$$

$$b = \frac{\sum_{k=2}^{n} x^{(1)}(k) \sum_{k=2}^{n} [z^{(1)}(k)]^2 - \sum_{k=2}^{n} z^{(1)}(k) \sum_{k=2}^{n} z^{(1)}(k) x^{(0)}(k)}{(n-1) \sum_{k=2}^{n} [z^{(1)}(k)]^2 - [\sum_{k=2}^{n} z^{(1)}(k)]^2} \tag{4}$$

The grey parameters can be substituted into Eq (1) to obtain the solution of the differential equation:

$$\hat{x}^{(1)}(k) = [x^{(0)}(1) - \frac{b}{a}]e^{-a(k-1)} + \frac{b}{a}, k = 1, 2, \cdots, n. \tag{5}$$

When $\hat{x}^{(1)} = x^{(0)}(1)$, the sequence one-order inverse-accumulated generating operation of reduction is obtained as:

$$\hat{x}^{(0)}(k) = \begin{cases} x^{(0)}(1) & , k = 1 \\ (1 - e^a)(x^{(0)}(1) - \frac{b}{a})e^{-a(k-1)} & , k \geq 2 \end{cases} \tag{6}$$

## 2.2 The wavelet residual modified grey forecasting model (WGM)

The accuracy of predictions made using GM(1,1), this study applied the WGM(1,1) approach to increase its prediction capabilities. Fig 1 shows the process of WGM(1,1), while the following illustration details the method used to establish the steps of WGM for the residual correction, which are detailed as follows.

(1)Definition of error:

$$\epsilon^{(0)}(j) = x^{(0)}(j) - \hat{x}^{(0)}(j), j = 2, 3, \cdots, n \tag{7}$$

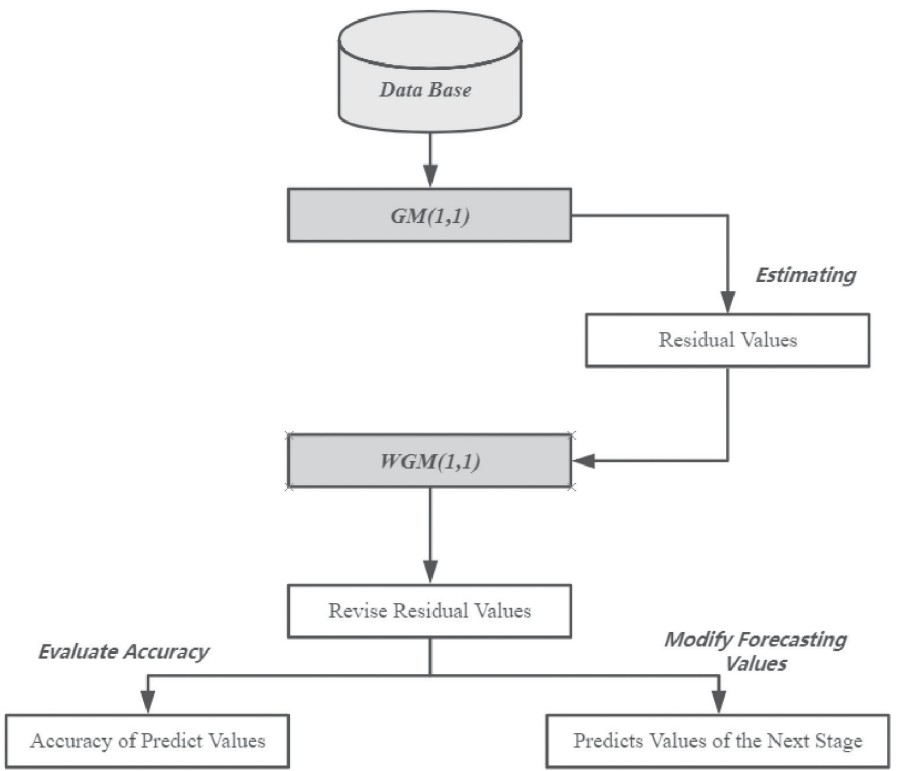

**Fig 1. Flow chart of WGM.**

We have

$$\epsilon^{(0)} = [\epsilon^{(0)}(2), \epsilon^{(0)}(3), \cdots, \epsilon^{(0)}(n)]^{T}.$$

(2)The wavelet basis is obtained by selecting the wavelet generating function

$$W_{l,i}(x) = 2^{-\frac{l}{2}}\Phi(2^{-l}x - i) = \frac{2}{\sqrt{3}}\pi^{\frac{1}{4}}[1 - (2^{-l}x - i)^2]e^{-\frac{1}{2}(2^{-l}x-i)^2}, (l \in Z, i = 2, 3, \cdots, n),$$

Then, we get the residual

$$\hat{\epsilon}^{(0)}(x) = \sum_{i=2}^{n} a_i W_{l,i}(x) \tag{8}$$

(3)Define the expression at a given two-dimensional data point $(j, \epsilon^{(0)}(j))$,

$$\Phi(a_2, a_3, \cdots, a_n) = \sum_{j=2}^{n}\{\sum_{i=2}^{n}[a_i W_{l,i}(j) - \epsilon^{(0)}(j)]\}^2 \tag{9}$$

(4)Use the least square method to determine the coefficient $a_2, a_3, \cdots, a_n$, then the following equations is solved

$$\sum_{k=2}^{n}\sum_{j=2}^{n}[W_{l,i}(j)W_{l,k}(j)a_k] = \sum_{k=2}^{n}W_{l,i}(j)\epsilon^{(0)}(j), i = 2, 3, \cdots, n \tag{10}$$

The matrix of the Eq (10) is expressed as

$$DC = W\epsilon^{(0)} \tag{11}$$

where

$$D = \begin{bmatrix} W_{l,i}(2)W_{l,2}(2) & W_{l,i}(2)W_{l,3}(2) & \cdots & W_{l,i}(2)W_{l,n}(2) \\ W_{l,i}(3)W_{l,2}(3) & W_{l,i}(3)W_{l,3}(3) & \cdots & W_{l,i}(3)W_{l,n}(3) \\ \vdots & \vdots & \vdots & \vdots \\ W_{l,i}(n)W_{l,2}(n) & W_{l,i}(n)W_{l,3}(n) & \cdots & W_{l,i}(n)W_{l,n}(n) \end{bmatrix},$$

$$C = [a_2, a_3, \cdots, a_n]^T,$$

$$W = [W_{l,i}(2), W_{l,i}(3), \cdots, W_{l,i}(n)]$$

Then, we have

$$C = (D^T D)^{-1} D^T W \epsilon^{(0)} \tag{12}$$

We have $a_2, a_3, \cdots, a_n$, So as to find the residual fitting function

$$\hat{\epsilon}^{(0)}(j) = \sum_{i=2}^{n} a_i W_{l,i}(j) \tag{13}$$

(5)Substituting the residual fitting function(13) into the error formula(7), we

$$\tilde{x}^{(0)}(j) = \hat{x}^{(0)}(j) + \hat{\epsilon}^{(0)}(j), (j \geq 2) \tag{14}$$

where

$$\hat{x}^{(0)}(j) = (1 - e^a)(x^{(0)}(1) - \frac{b}{a})e^{-a(j-1)}, (j = 2, 3, \cdots, n)$$

The original value prediction function is:

$$\tilde{x}^{(0)}(j) = \begin{cases} x^{(0)}(1) & , j = 1 \\ (1 - e^a)\left(x^{(0)}(1) - \frac{b}{a}\right)e^{-a(j-1)} + \sum_{i=2}^{n} a_i W_{l,i}(j) & , j \geq 2 \end{cases} \tag{15}$$

## 2.3 Evelute accuracy of prediction

The accuray of the grey prediction model is usually checked by the posterior difference method. The model accuracy is evaluated by the mean square error ratio and the small error probability. The smaller the mean square error ratio and the smaller the probability of small errors, the higher the accuracy of the prediction model, The basic method is as follows: assuming $X^{(0)}$ for the original sequence, $\hat{x}^{(0)}$ for the GM(1,1) model simulation sequence, $\epsilon^{(0)}$ for the

residual series, then

$$\bar{x} = \frac{1}{n}\sum_{k=1}^{n} x^{(0)}(k),$$

$$S_1^2 = \frac{1}{n}\sum_{k=1}^{n} [x^{(0)}(k) - \bar{x}]^2,$$

Respectively, the mean of $X^{(0)}$, variance;

$$\epsilon^{(0)}(k) = x^{(0)}(k) - \hat{x}^{(0)}(k),$$

$$\bar{\epsilon} = \frac{1}{n}\sum_{k=1}^{n} \epsilon^{(0)}(k),$$

$$S_2^2 = \frac{1}{n}\sum_{k=1}^{n} [\epsilon^{(0)}(k) - \bar{\epsilon}]^2,$$

$$K = \frac{S_2}{S_1}.$$

Where $\bar{\epsilon}$ is the mean of resdual $\epsilon^{(0)}(k)$,$S_2^2$ is the residual variance, and $K$ is the mean square deviation. The mean square error ratio $K$ and the small error probability $P$ are calculated from $K = \frac{S_2}{S_1}$ and $P = p(\mid \epsilon(k) - \bar{\epsilon} \mid < 0.6745S_1)$. The accuracy of the model is shown in Table 1.

In addition to the evaluation of the posterior error index, there are the following evaluation methods:

(1)Average residual

$$\frac{1}{n}\sum_{k=1}^{n} [x^{(0)}(k) - \tilde{x}^{(0)}(k)],$$

(2)Average relative errore

$$\frac{1}{n}\sum_{k-1}^{n} [\frac{\mid\mid x^{(0)}(k) - \tilde{x}^{(0)}(k)\mid}{x^{(0)}(k)}],$$

(3)Mean absolute error

$$MAE = \frac{1}{n}\sum_{k=1}^{n} \mid[x^{(0)}(k) - \tilde{x}^{(0)}(k)]\mid,$$

**Table 1. The reference table of precision check grade.**

| Modelaccuracylevel | Small probability of error P | Mean square deviation ratio K |
|---|---|---|
| Level1(good) | $P \geq 0.95$ | $K \leq 0.35$ |
| Level2(qualified) | $0.80 \leq P < 0.95$ | $0.35 < K \leq 0.50$ |
| Level3(barely) | $0.70 \leq P < 0.80$ | $0.50 < K \leq 0.65$ |
| Level4(unqualified) | $P < 0.70$ | $K > 0.65$ |

(4)Mean square error

$$MSE = \frac{1}{n}\sum_{k=1}^{n}[x^{(0)}(k) - \tilde{x}^{(0)}(k)]^2.$$

## 3.Application experiment analysis

The data of Example 1 and Example 2 analysis WGM model prediction accuracy. The experiments were done using Matlab2015b software.

### 3.1 Simulation value analysis

**Example 1:** The 10-day stock price of PetroChina in May 2018 was used as the original data for the experiment. PetroChina stock price data comes from literature [31].

The experimental data calculates that $a = 0.00122$, $b = 17.60864$ in the mean model, then the calculation formula of the mean model simulation value is:

$$\hat{x}^{(0)}(j) = \begin{cases} 17.91 & ,j = 1 \\ -14415.36(1 - e^{0.00122})e^{-0.00122j+0.00122} & ,j \geq 2 \end{cases}$$

Coefficients of Wavelet Base Residual Fitting $[a_1, a_2, \cdots, a_n]$, $a_i = [0, -1556.125479137, 1813.934889951, -323.708557033, -299.425973673, -363.424104224, 1615.482710942, -2666.599778900, 2939.837180356, -1677.799794262]$, the simulation value of Wavelet Residual Grey Forecasting Model is:

$$\tilde{x}^{(0)}(j) = \begin{cases} 17.91 & ,j = 1 \\ -14415.36(1 - e^{0.00122})e^{-0.00122j+0.00122} + \sum_{i=2}^{n} a_i W_{l,i}(j) & ,j \geq 2 \end{cases}$$

It can be seen from the above Table 2 that the mean residual error and the mean relative error of the residual correction model of the new method are $2.5602 \times 10^{-9}$ and $5.235025317962417 \times 10^{-14}$, respectively, while the FGM(1, 1) models are 0.130 and 0.849031398741481, respectively. It can be seen from this order of magnitude that the residual correction model of the new method has higher simulation accuracy than the FGM(1,1) model and the GM(1,1), which indicates that the residual correction model of the new method has the best simulation effect.

### 3.2 Simulation value analysis

**Example 2:** The experiment was conducted with the original data of the number of tourists visiting Taiwan during August 1, 2006–July 31, 2007. The data of the number of tourists visiting Taiwan comes from literature [29].

The experimental data calculates that $a = 0.00491$, $b = 179544.95413$ in the mean model is calculated from the experimental data, then the calculation formula for the simulation value of the mean model is:

$$\hat{x}^{(0)}(j) = \begin{cases} 193510 & ,j = 1 \\ -36373496(1 - e^{0.00491})e^{-0.00491j+0.00491} & ,j \geq 2 \end{cases}$$

Coefficients of Wavelet Base Residual Fitting $[a_1, a_2, \cdots, a_n]$, $a_i = [0, -93.9267590491499, -177.300688508506, 512.222285037394, -689.422488712347, 726.337872801881,$

**Table 2. Traditional GM(1,1) simulation of PetroChina stock price.**

| Date | j | Initial data | GM(1,1) | | | FGM(1,1) | | | WGM(1,1) | | |
|---|---|---|---|---|---|---|---|---|---|---|---|
| | | | Data1 | Residual | Relative error | Data2 | Residual | Relative error | Data3 | Residual | Relative error |
| May 09 | 1 | 17.91 | 17.910000 | 0 | 0 | 17.910000 | 0 | 0 | 17.9100000000000 | 0 | 0 |
| May 12 | 2 | 17.89 | 17.576153 | 0.313846 | 0.017543 | 17.884175 | 0.005824 | 0.000325 | 17.8900002615374 | $-2.61537433488002 \times 10^{-7}$ | $1.46191969529347 \times 10^{-8}$ |
| May 13 | 3 | 17.39 | 17.554779 | -0.164779 | 0.009475 | 17.472043 | -0.082043 | 0.004717 | 17.3900000772547 | $-7.72547394944922 \times 10^{-8}$ | $4.44248070698633 \times 10^{-9}$ |
| May 14 | 4 | 17.65 | 17.533430 | 0.116569 | 0.006604 | 17.501633 | 0.148366 | 0.008406 | 17.6500000284613 | $-2.84613363987773 \times 10^{-8}$ | $1.61254030587974 \times 10^{-9}$ |
| May 16 | 5 | 17.61 | 17.512108 | 0.097891 | 0.005558 | 17.806795 | -0.196795 | 0.011175 | 17.6100000243747 | $-2.43746569594805 \times 10^{-8}$ | $1.38413724926068 \times 10^{-9}$ |
| May 19 | 6 | 17.35 | 17.490811 | -0.140811 | 0.008115 | 17.128512 | 0.221487 | 0.012765 | 17.3500000001375 | $-1.37461597660149 \times 10^{-10}$ | $7.92285865476363 \times 10^{-12}$ |
| May 20 | 7 | 16.75 | 17.469541 | -0.719541 | 0.042957 | 16.969464 | -0.219464 | 0.013102 | 16.7500000057193 | $-5.71927927239813 \times 10^{-9}$ | $3.41449508799888 \times 10^{-10}$ |
| May 21 | 8 | 17.86 | 17.448296 | 0.411703 | 0.023051 | 17.669029 | 0.190970 | 0.010692 | 17.8600000086396 | $-8.63959570551742 \times 10^{-9}$ | $4.83739961115197 \times 10^{-10}$ |
| May 22 | 9 | 17.46 | 17.427077 | 0.032922 | 0.001885 | 17.599443 | -0.139443 | 0.007986 | 17.4600000320697 | $-3.20696500466511 \times 10^{-8}$ | $1.83674971630304 \times 10^{-9}$ |
| May 23 | 10 | 17.66 | 17.405884 | 0.254115 | 0.014389 | 17.497393 | 0.162606 | 0.009207 | 17.6600000154387 | $-1.54386903261639 \times 10^{-8}$ | $8.74218025264091 \times 10^{-10}$ |
| Average Residual | | | 0.0202 | | | 0.0092 | | | $-9.1292 \times 10^{-8}$ | | |
| Average Relative Error | | | 0.0130 | | | 0.0078 | | | $2.5602 \times 10^{-9}$ | | |
| Mean Absolute Error | | | 0.2252 | | | 0.1367 | | | $9.1292 \times 10^{-8}$ | | |
| Mean Square Error | | | 0.9922 | | | 0.0247 | | | $7.714848273949280 \times 10^{-15}$ | | |
| Posterior Error Ratio K | | | 0.849031398741481 | | | 0.227876033693310 | | | $5.235025317962417 \times 10^{-14}$ | | |
| Small Error Probability P | | | 1 | | | 1 | | | 1 | | |

−242.072495924380, −267.280726217092,298.150674304123, −243.004450579142, 328.272731679794, −238.449423788114], the simulation value of Wavelet Residual Grey Forecasting Model is:

$$\tilde{x}^{(0)}(j) = \begin{cases} 193510 & ,j = 1 \\ -36373496(1 - e^{0.00491})e^{-0.00491j+0.00491} + \sum_{i=2}^{n} a_i W_{l,i}(j) & ,j \geq 2 \end{cases}$$

As can be seen from the above Table 3, the mean residual error and mean relative error of the new residual correction model method are $1.1008 \times 10^{-7}$ and $2.451254681202029 \times 10^{-12}$, respectively, while the FGM(1,1) model is 0.0296 and 0.886095333,respectively. From this order of magnitude, it can be seen that the residual correction model of the new method has higher simulation accuracy than the FGM(1,1) model and the GM(1,1), which indicates that the residual correction model of the new method has better simulation effect.

### 3.3 WGM model simulation diagram analysis

The statistics of the Tables 2 and 3 show the actual number,forecast number,residual error, average residual,average relative error,mean absolute error, mean square error and posterior

**Table 3. The number of tourist visits to Taiwan.**

| Year | Month | j | Initial data | GM(1,1) | | | FGM(1,1) | | | WGM(1,1) | | |
|---|---|---|---|---|---|---|---|---|---|---|---|---|
| | | | | Data1 | Residual | Relative error | Data2 | Residual | Relative error | Data3 | Residual | Relative error |
| 2006 | Nov | 1 | 193510 | 193510 | 0 | 0 | 193510 | 0 | 0 | 193509.982251622 | 0.0177483777515590 | 0 |
| | Dec | 2 | 190390 | 178156 | 12234 | 0.06425 | 198492 | -8102 | 0.042558565050647342 | 190390.091995750 | -0.0919957496225834 | 4.83196331858729×10−7 |
| 2007 | Jan | 3 | 173730 | 177284 | -3554 | -0.02045 | 165156 | 8574 | 0.049348411937138 0 | 173730.039353058 | -0.039353057949495 | 2.26518494070969×10−7 |
| | Feb | 4 | 147280 | 176415 | -29135 | -0.19782 | 155629 | -8349 | 0.0566900886903458 | 147280.006809756 | -0.006809756421716 88 | 4.62368035151880×10−8 |
| | Mar | 5 | 210610 | 175551 | 35059 | 0.16646 | 203161 | 7449 | 0.0353683157044412 | 210610.001628656 | -0.0016286559985019 3 | 7.73304210864597×10−8 |
| | Apr | 6 | 175030 | 174691 | 339 | 0.00193 | 180975 | -5945 | 0.0339659395653226 | 175029.987908301 | 0.0120916987361852 | 6.90835784504668×10−8 |
| | May | 7 | 171540 | 173835 | -2295 | -0.01338 | 167580 | 3960 | 0.0230824446695148 | 171539.997623990 | 0.0023760096228215 8 | 1.38510529487092×10−8 |
| | Jun | 8 | 175540 | 172984 | 2556 | 0.01455 | 177193 | -1653 | 0.00941828930742561 | 175540.004259859 | -0.0042598586296662 7 | 2.42671677661289×10−8 |
| | Jul | 9 | 148140 | 172136 | -23996 | -0.16198 | 148926 | -786 | 0.00531206283732469 | 148140.012040685 | -0.0120406854839530 | 8.12790973670378×10−8 |
| | Aug | 10 | 171200 | 171293 | -93 | -0.00054 | 168036 | 3164 | 0.0184777580365000 | 171200.017515752 | -0.0175157518824562 | 1.02311634827431×10−7 |
| | Sep | 11 | 177530 | 170454 | 7076 | 0.03985 | 182813 | -5283 | 0.0297615960249348 | 177530.018872230 | -0.0188722294988111 | 1.06304452761849×10−7 |
| | Oct | 12 | 181170 | 169619 | 11551 | 0.06375 | 171906 | 9264 | 0.0511296325629720 | 181170.029011169 | -0.0290111690410413 | 1.60132301380147×10−7 |
| Average Residual | | | | | 811.3591 | | | 190.6199 | | | -0.0158 | |
| Average Relative Error | | | | | 0.0621 | | | 0.0296 | | | 1.1008×10−7 | |
| Mean Absolute Error | | | | | 10657 | | | 5210.8 | | | 0.0211 | |
| Mean Square Error | | | | | 250960053.7199431 | | | 36871406.11542063 | | | 9.9004133878328 50×10−4 | |
| Posterior error ratio K | | | | | 0.886095333133232 | | | 0.130400141967458 | | | 2.45125468120202 9×10−12 | |
| Small error Probability P | | | | | 1 | | | 1 | | | 1 | |

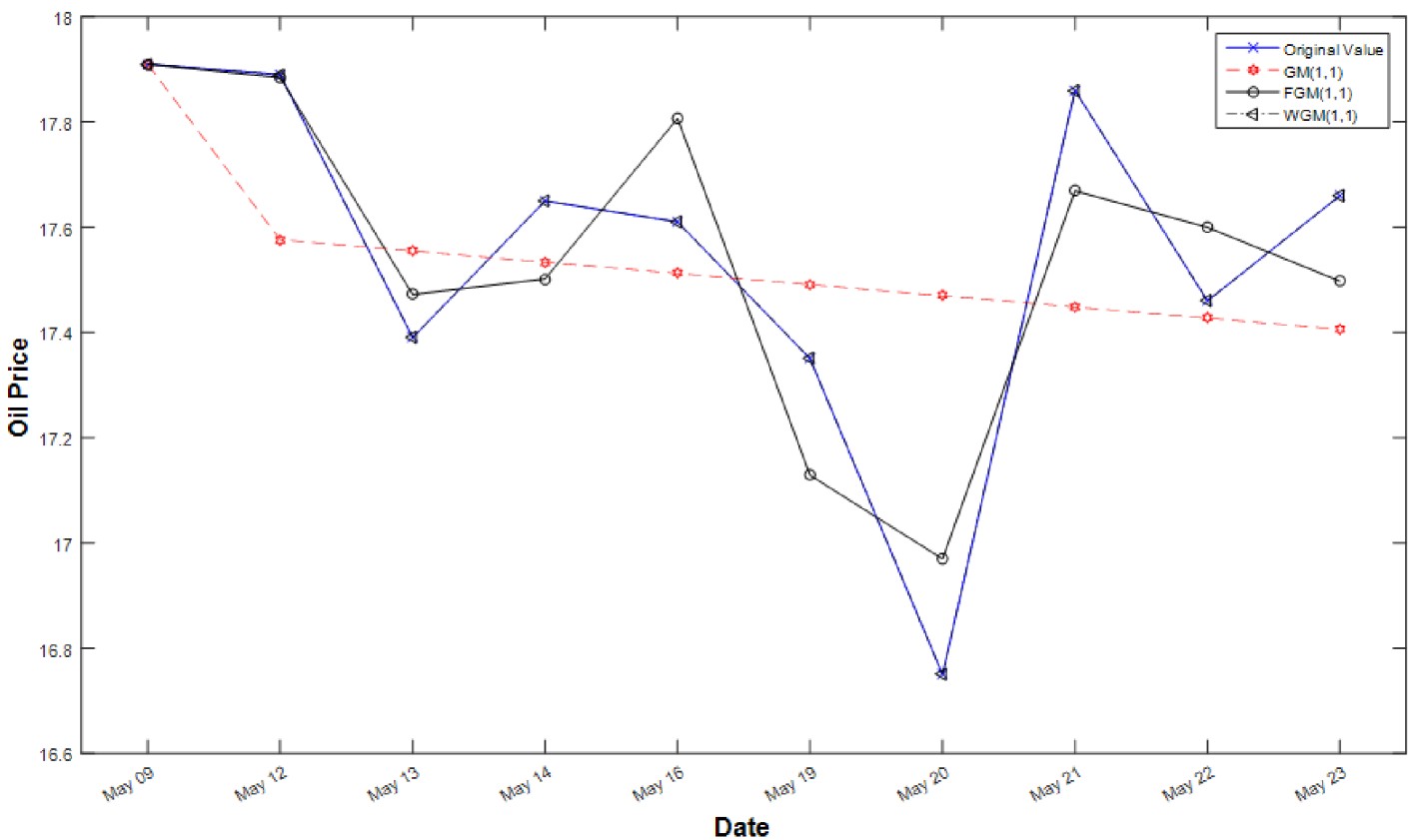

**Fig 2. Line graphs of simulated values after processing of the three models.**

error by GM(1,1) and FGM(1,1) and WGM(1,1). The evaluation indicators displayed from the two different sets of data show that the evaluation indicators of the WGM method constructed in this study are far superior to the evaluation indicators corresponding to the GM method and the FGM method. It shows that the simulation accuracy of the WGM method is higher than that of the GM method and the FGM method, and the fit of the WGM method to the random fluctuation data is better, which increases the application range of the grey system.

The statistics of the Tables 2 and 3 indicate the short-term efficiency of the WGM. Figs 2 and 3 compare actual value, GM forecasting value and FGM forecasting value and WGM forecasting value. The results show that the cures of GM appear flat and FGM and WGM follows undulating appearance, but WGM is more effective than FGM in predicting stochastic volatility data. From the results, this study develops an accuracy forecasting model for improving the effectiveness of PteroChina stock price and taiwan's tourism demand in making short-term and stochastic volatility predictions.

## 4.Summary

From the experimental analysis, the WGM has better simulation effect and higher prediction accuracy than the GM and FGM. The prediction effect of the WGM established in this paper is better than the FGM established in literature [29], and the simulation accuracy of the WGM is higher.

According to the residual analysis and research of the grey model (GM(1,1)), a new method residual correction prediction model(WGM) is established in this paper. This study of WGM

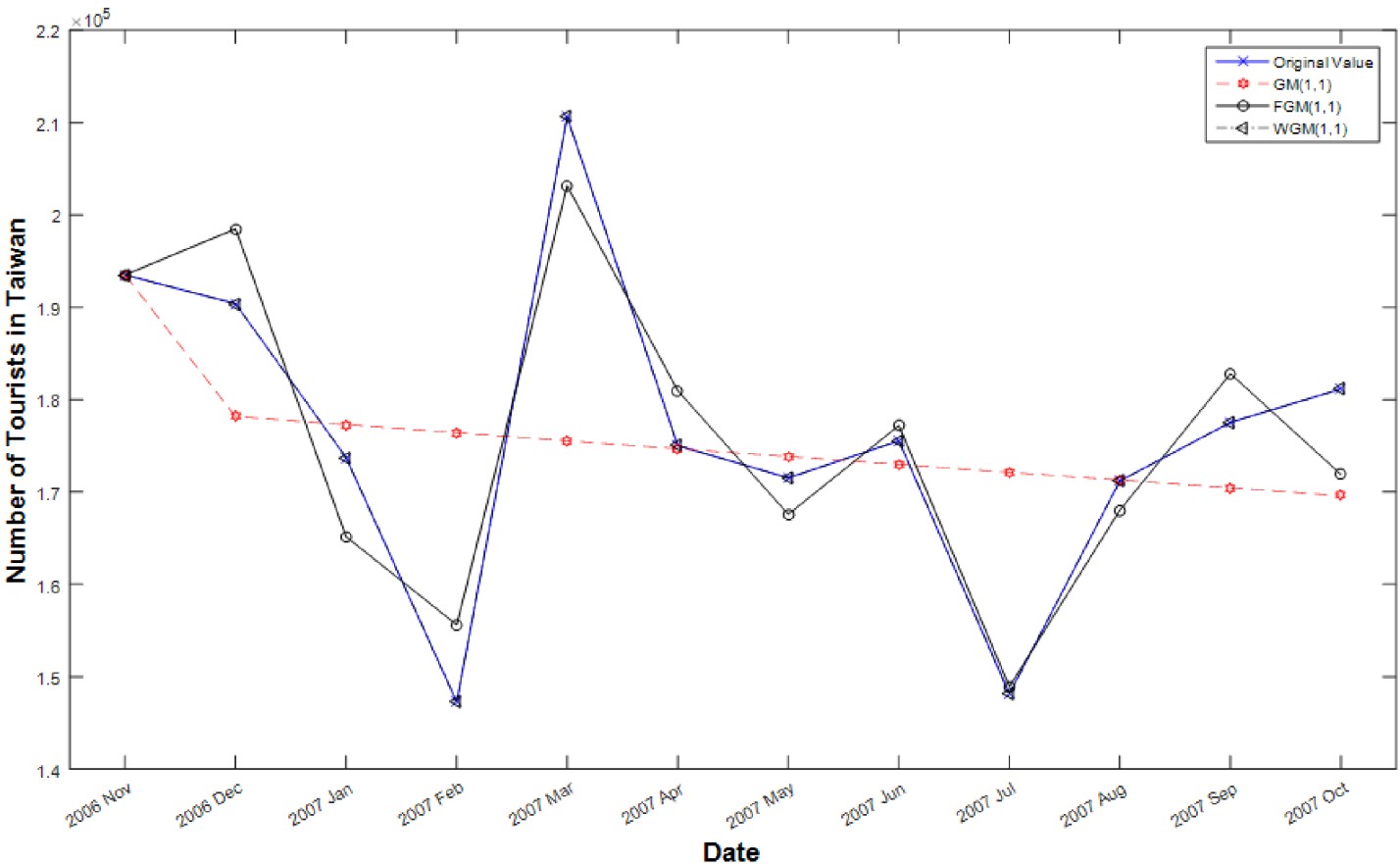

**Fig 3. Line graphs of simulated values after processing of the three models.**

makes the following contributions: 1)combing the grey forecasting and wavelet series model to refine the forecasting effectiveness for the stochastic volatility data; 2)proving an effective method for the application; 3)improving the accuracy of short-term forecasting in case involving sample data with significant fluctuations.

## Supporting information

**S1 File.**
(ZIP)

**S1 Text.**
(TXT)

## Acknowledgments

Thanks are due to Chenzhou Deng for assistance with the experiments and to Tian Ding for valuable discussion.

## Author Contributions

**Conceptualization:** Lifang Xiao.

**Formal analysis:** Xiangyang Chen.

**Software:** Hao Wang.

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
