## [Decision Letter · Decision Letter 0]

3 Mar 2021

PONE-D-20-39606

Calculation and Realization of New Method Grey Residual Error Correction Model

PLOS ONE

Dear Dr. Xiao,

Thank you for submitting your manuscript to PLOS ONE. After careful consideration, we feel that it has merit but does not fully meet PLOS ONE’s publication criteria as it currently stands. Therefore, we invite you to submit a revised version of the manuscript that addresses the points raised during the review process.

We look forward to receiving your revised manuscript.

Kind regards,

Dragan Pamucar

Academic Editor

PLOS ONE

Journal Requirements:

2. Please amend your list of authors on the manuscript to ensure that each author is linked to an affiliation. Authors’ affiliations should reflect the institution where the work was done (if authors moved subsequently, you can also list the new affiliation stating “current affiliation:….” as necessary).

3. Please ensure that you refer to Figures (1-4) in your text as, if accepted, production will need this reference to link the reader to the figure.

4. We note you have included a table to which you do not refer in the text of your manuscript. Please ensure that you refer to Tables (1-4) in your text; if accepted, production will need this reference to link the reader to the Table

Reviewers' comments:

Reviewer's Responses to Questions

**Comments to the Author**

1. Is the manuscript technically sound, and do the data support the conclusions?

Reviewer #1: Yes

Reviewer #2: Partly

2. Has the statistical analysis been performed appropriately and rigorously? 

Reviewer #1: Yes

Reviewer #2: I Don't Know

3. Have the authors made all data underlying the findings in their manuscript fully available?

Reviewer #1: Yes

Reviewer #2: Yes

4. Is the manuscript presented in an intelligible fashion and written in standard English?

Reviewer #1: Yes

Reviewer #2: No

5. Review Comments to the Author

Reviewer #1: The paper proposed a method to improve the prediction accuracy of the grey model based on the residual error correction method, and this new method (i.e., the improved GM method) was applied in three experimental analyses from the perspective of simulating the stock price of PetroChina, the population of US and the per capita energy consumption of China, respectively. Moreover, evaluation index including average residual, mean relative error and posterior difference ratio were selected, when compared the traditional GM method with the new method established by the residual correction simulation method. The effort of the authors is greatly appreciated, but in my opinion, Partial content needs to be improved, my comments are as follows:

1. It is obvious that the improved GM method is capable of addressing the issue of few samples and poor information in economy, management and engineering technology compared to the GM method. However, high requirements on the input data were not emphatic in this paper, considering the input data such as the stock price, the population and the per capita energy consumption present similar linear features and have the same sign. However, some input data present waveform features, or even highly non-linear features in reality, which was neglected in this paper. Therefore, input data selected in three experimental analyses should be representative and different, in other words, it is better to have three different characteristics of data rather than the same characteristics of data. As a consequence, emphasize the characteristics of the input data and change the type of the experimental analysis are necessary.

2. Similar to other prediction methods, the GM method also bears the limitations. Hence, the issue of improving the accuracy has received considerable critical attention. Nevertheless, literature and review involving introducing the advances in the GM method is lacking, especially the research progress of foreign scholars in this field. In addition, residual GM method mentioned is a common method to amend the traditional GM method and have been adopted in many aspects. Therefore, the innovation of this new improved GM method is insufficient. As a contrast, some other improved GM methods need to be introduced to highlight the advantages of the improved GM method in this paper.

3. It is worth mentioning that evaluation index in this paper have no explicit formula expressions. Except for average residual, mean relative error (MRE) and posterior difference ratio mentioned in this paper, other indicators such as mean absolute error (MAE), mean square error (MSE) and mean absolute percentage error (MAPE) should be taken into consideration. Moreover, the flow chart of constructing the improved GM method is necessary, given the complexity of the formula derivation.

Given the above three comments, I advise the paper need major changes, adding some important references bibliography at the same time.

Reviewer #2: The paper is suggested to be accepted with major revision. The authors are suggested to follow up the below comments.

1)There is still room to improve English writing. Please improve English and engage native proof-reader if available.

There are too many grammatical errors, such as too many sentences without subject.

“According to the characteristics of the original data, improve the gray prediction algorithm, optimize the background values, optimize the initial conditions, etc.”

And please try more passive sentences.

2) It is beneficial to cite other related publications in Plus One to reflect the relevance of your submission.

3) The amount of data is too small, it is limited for the examples of experimental analysis.

4) It will be helpful to add a flow-process diagram for describing the new residual correction grey model.

5) By the example data analysis, it can be found that the fitting accuracy and prediction effect of this method are better than the traditional GM(1,1) model. But how does it compare with other improved models?

6. PLOS authors have the option to publish the peer review history of their article (what does this mean?). If published, this will include your full peer review and any attached files.

Reviewer #1: No

Reviewer #2: No

---

## [Author Response · Author response to Decision Letter 0]

20 May 2021

Dear reviewer1, 

 I am very grateful to you for taking the time to read my article,and putting forward a lot of valuable suggestions for amendments. According to your suggestions, I will answer one by one as follows:

Question1: It is obvious that the improved GM method is capable of addressing the issue of few samples and poor information in economy, management and engineering technology compared to the GM method. However, high requirements on the input data were not emphatic in this paper, considering the input data such as the stock price, the population and the per capita energy consumption present similar linear features and have the same sign. However, some input data present waveform features, or even highly non-linear features in reality, which was neglected in this paper. Therefore, input data selected in three experimental analyses should be representative and different, in other words, it is better to have three different characteristics of data rather than the same characteristics of data. As a consequence, emphasize the characteristics of the input data and change the type of the experimental analysis are necessary.

Answer1：Thank the reviewer for their valuable suggestions. Based on the data characteristics question raised by the reviewer, this article emphasizes that the research method is based on the data of stochastic volatility characteristics. The data of the closing stock price of PetroChina and the number of tourists in Taiwan is experimented, and the study finds that the WGM method proposed in this article is superior to the FGM (1, 1) method in the literature [29] and GM (1, 1) method.

Question2:Similar to other prediction methods, the GM method also bears the limitations. Hence, the issue of improving the accuracy has received considerable critical attention. Nevertheless, literature and review involving introducing the advances in the GM method is lacking, especially the research progress of foreign scholars in this field. In addition, residual GM method mentioned is a common method to amend the traditional GM method and have been adopted in many aspects. Therefore, the innovation of this new improved GM method is insufficient. As a contrast, some other improved GM methods need to be introduced to highlight the advantages of the improved GM method in this paper.

Answer2：Thank the reviewer for their valuable suggestions. This article has added literature and reviews of the progress of the GM method in the introduction, including the addition of some foreign scholars' research progress in this field. In addition, in improving the innovation of the GM method, this paper proposes to use the wavelet method to fit the residuals to correct the error caused by the GM method. The combination of the wavelet method and the GM method makes the fitting effect very good. This is the innovative point of this article. 

Question3:It is worth mentioning that evaluation index in this paper have no explicit formula expressions. Except for average residual, mean relative error (MRE) and posterior difference ratio mentioned in this paper, other indicators such as mean absolute error (MAE), mean square error (MSE) and mean absolute percentage error (MAPE) should be taken into consideration. Moreover, the flow chart of constructing the improved GM method is necessary, given the complexity of the formula derivation.

Answer3：Thank the reviewer for their valuable suggestions. This article has added the formula expression of the evaluation index, and in addition to the average residual error, the average relative error and the posterior error, the average absolute error and average square error of the evaluation index is also added. This article has added a flow chart for improving the GM method.

Given the above three comments, I advise the paper need major changes, adding some important references bibliography at the same time.

Answer：Thank the reviewers for their valuable suggestions. This article has been significantly revised. First, the wavelet method is used to fit the residuals, and the residual prediction function is established. Then, the residual prediction function is used to correct the errors caused by the GM method, and the WGM model is established. At the same time, this article has added the books and documents written by the founders of the GM method ,Professor Deng Julong,etc., as well as some documents by foreign scholars.

Kind regards,

 Lifang Xiao

Dear reviewer2, 

 I am very grateful to you for taking the time to read my article,and putting forward a lot of valuable suggestions for amendments. According to your suggestions, I will answer one by one as follows:

Question1：There is still room to improve English writing. Please improve English and engage native proof-reader if available.

There are too many grammatical errors, such as too many sentences without subject.“According to the characteristics of the original data, improve the gray prediction algorithm, optimize the background values, optimize the initial conditions, etc.”And please try more passive sentences.

Answer1：Thank the reviewer for their valuable suggestions. The revised manuscript has been revised for grammatical problems. The revised manuscript of this article has emphasized the stochastic volatility characteristics of the data, and the methods of optimizing the background value and optimizing the initial conditions are not effective in fitting the stochastic volatility characteristics. This article is modified to use the wavelet method to fit the residuals and the residual prediction function is established, and the residuals prediction function is used to correct the error caused by the GM method, and the WGM model is established. The WGM method achieves a very good fitting effect.

Question2： It is beneficial to cite other related publications in Plus One to reflect the relevance of your submission.

Answer2：Thank the reviewer for their valuable suggestions. This article has added relevant documents of PlosOne.

Question3：The amount of data is too small, it is limited for the examples of experimental analysis.

Answer3：Thank the reviewer for their valuable suggestions. The biggest advantage of the GM method is to predict unknown information based on a small amount of information. This is different from the big data method.

Question4： It will be helpful to add a flow-process diagram for describing the new residual correction grey model.

Answer4：Thank the reviewer for their valuable suggestions. The revised manuscript has added a flowchart of the improving GM method.

Question5： By the example data analysis, it can be found that the fitting accuracy and prediction effect of this method are better than the traditional GM(1,1) model. But how does it compare with other improved models?

Answer5：Thank the reviewer for their valuable suggestions. The revised paper mentions the FGM method in the literature [29]. The new WGM method studied in this article is compared with the FGM method and the traditional GM method. The experiment shows that the fitting accuracy and fitting effect of the WGM method are both Better than FGM method and GM method.

Kind regards,

 Lifang Xiao

---

## [Decision Letter · Decision Letter 1]

22 Jun 2021

Calculation and Realization of New Method Grey Residual Error Correction Model

PONE-D-20-39606R1

Dear Dr. Xiao,

We’re pleased to inform you that your manuscript has been judged scientifically suitable for publication and will be formally accepted for publication once it meets all outstanding technical requirements.

Kind regards,

Dragan Pamucar

Academic Editor

PLOS ONE

Additional Editor Comments (optional):

Reviewers' comments:

Reviewer's Responses to Questions

**Comments to the Author**

1. If the authors have adequately addressed your comments raised in a previous round of review and you feel that this manuscript is now acceptable for publication, you may indicate that here to bypass the “Comments to the Author” section, enter your conflict of interest statement in the “Confidential to Editor” section, and submit your "Accept" recommendation.

Reviewer #1: All comments have been addressed

Reviewer #2: All comments have been addressed

2. Is the manuscript technically sound, and do the data support the conclusions?

Reviewer #1: Yes

Reviewer #2: Yes

3. Has the statistical analysis been performed appropriately and rigorously? 

Reviewer #1: Yes

Reviewer #2: Yes

4. Have the authors made all data underlying the findings in their manuscript fully available?

Reviewer #1: (No Response)

Reviewer #2: Yes

5. Is the manuscript presented in an intelligible fashion and written in standard English?

Reviewer #1: Yes

Reviewer #2: Yes

6. Review Comments to the Author

Reviewer #1: The manuscript is acceptable and could be published in the journal PLOS ONE. Comments on the lack of a "discussion" point, would improve the readability of the manuscript, but their disregard by the authors does not negatively affect on the quality of the manuscript.

Reviewer #2: The authors have adequately addressed your comments raised in a previous round of review，and I feel that this manuscript is now acceptable for publication.

7. PLOS authors have the option to publish the peer review history of their article (what does this mean?). If published, this will include your full peer review and any attached files.

Reviewer #1: No

Reviewer #2: No

---

## [Editor Report · Acceptance letter]

8 Jul 2021

PONE-D-20-39606R1 

Calculation and Realization of New Method Grey Residual Error Correction Model 

Dear Dr. Xiao:

I'm pleased to inform you that your manuscript has been deemed suitable for publication in PLOS ONE. Congratulations! Your manuscript is now with our production department. 

Kind regards, 

on behalf of

Dr. Dragan Pamucar 

Academic Editor

PLOS ONE